# Risk of bias in prognostic models of hospital-induced delirium for medical-surgical units: A systematic review

Urszula A. Snigurska[1]*, Yiyang Liu[2], Sarah E. Ser[2], Tamara G. R. Macieira[1], Margaret Ansell[3], David Lindberg[4], Mattia Prosperi[2], Ragnhildur I. Bjarnadottir[1], Robert J. Lucero[1,5]

1 Department of Family, Community, and Health Systems Science, College of Nursing, University of Florida, Gainesville, FL, United States of America, 2 Department of Epidemiology, College of Public Health and Health Professions and College of Medicine, University of Florida, Gainesville, FL, United States of America, 3 Health Science Center Libraries, George A. Smathers Libraries, University of Florida, Gainesville, FL, United States of America, 4 Department of Statistics, College of Liberal Arts and Sciences, University of Florida, Gainesville, FL, United States of America, 5 School of Nursing, University of California Los Angeles, Los Angeles, CA, United States of America

* usnigurska@ufl.edu

## Abstract

### Purpose

The purpose of this systematic review was to assess risk of bias in existing prognostic models of hospital-induced delirium for medical-surgical units.

### Methods

APA PsycInfo, CINAHL, MEDLINE, and Web of Science Core Collection were searched on July 8, 2022, to identify original studies which developed and validated prognostic models of hospital-induced delirium for adult patients who were hospitalized in medical-surgical units. The Checklist for Critical Appraisal and Data Extraction for Systematic Reviews of Prediction Modelling Studies was used for data extraction. The Prediction Model Risk of Bias Assessment Tool was used to assess risk of bias. Risk of bias was assessed across four domains: participants, predictors, outcome, and analysis.

### Results

Thirteen studies were included in the qualitative synthesis, including ten model development and validation studies and three model validation only studies. The methods in all of the studies were rated to be at high overall risk of bias. The methods of statistical analysis were the greatest source of bias. External validity of models in the included studies was tested at low levels of transportability.

### Conclusions

Our findings highlight the ongoing scientific challenge of developing a valid prognostic model of hospital-induced delirium for medical-surgical units to tailor preventive

**Data Availability Statement:** All relevant data are within the paper and its Supporting Information files.

**Funding:** This research was supported by the National Institute on Aging (https://www.nia.nih.gov/), R33 AG062884; RIB and RJL, Multiple Principal Investigators. The funders had no role in study design, data collection and analysis, decision to publish, or preparation of the manuscript.

**Competing interests:** The authors have declared that no competing interests exist.

interventions to patients who are at high risk of this iatrogenic condition. With limited knowledge about generalizable prognosis of hospital-induced delirium in medical-surgical units, existing prognostic models should be used with caution when creating clinical practice policies. Future research protocols must include robust study designs which take into account the perspectives of clinicians to identify and validate risk factors of hospital-induced delirium for accurate and generalizable prognosis in medical-surgical units.

## Introduction

Every year delirium complicates hospital stays for greater than 2.3 million adults of 65 years and older who are hospitalized in the United States [1]. The financial burden of delirium among hospitalized older adults on the health care system in the United States ranges from $38 to $152 billion every year [1]. Delirium refers to an acute neurocognitive syndrome which is characterized by disturbance in attention and awareness with fluctuating intensity [2]. Older adults are at a higher risk of hospital-induced delirium because they typically have more predisposing factors [3]. Gibb et al. (2020) reports an estimated 23% occurrence of delirium among hospitalized older adults [4]. The development of hospital-induced delirium is associated with subsequent cognitive and functional decline [5]. Moreover, the risk of death among older adults with hospital-induced delirium is three times as high as among older adults without hospital-induced delirium [6].

Prognostic models of hospital-induced delirium can identify patients who are at high risk of developing delirium and inform the design and implementation of tailored preventive interventions. For example, the Hospital Elder Life Program, a widely adopted intervention which is based on a prognostic model, has been successful in the primary prevention of hospital-induced delirium among older adults [7–9]. According to Baker and Gerdin (2017), "good" predictive performance (i.e., discrimination and calibration) is a prerequisite for clinical usefulness of a prognostic model [10]. However, predictive performance can be distorted due to bias during the development and/or validation of a model. Model development and validation studies need to be assessed for risk of bias to evaluate the validity of prognostic models. This is a critical step before they can be implemented in clinical practice.

Prior systematic reviews have evaluated existing prognostic models of hospital-induced delirium [11–16]. However, all of these systematic reviews included models which were developed for intensive care units. Meanwhile, patients who are hospitalized in medical-surgical units are also at risk of developing hospital-induced delirium. This is especially the case for older adults [17]. Significant risk factors for patients who are hospitalized in medical-surgical units may be different than the risk factors for patients who are hospitalized in intensive care units. The purpose of our systematic review was to assess risk of bias in existing prognostic models of hospital-induced delirium for medical-surgical units.

## Methods

This systematic review is based on the protocol which has been registered in the International Prospective Register of Systematic Reviews PROSPERO under the registration number CRD42020218635. This manuscript adheres with the Preferred Reporting Items for Systematic Reviews and Meta-Analyses (PRISMA) 2020 guidelines [18].

## Data sources

The search for relevant literature was conducted on July 8, 2022, and spanned four databases for health sciences: Cumulative Index to Nursing and Allied Health Literature (CINAHL) via EBSCOHost, MEDLINE via PubMed, APA PsycInfo via EBSCOHost, and Web of Science Core Collection.

## Search strategy

The search strategy for each database was designed to retrieve literature which included the concepts of delirium (group 1) in the hospital setting (group 2) and referred to models and their development and validation (group 3). The search terms for each group were identified with the help of our Nursing and Consumer Health Liaison Librarian. Both controlled vocabulary and keywords were utilized. The full syntax for each database can be found in the supplementary material (S1 File). An English language filter was applied to the search. No date limits were applied.

## Eligibility criteria

Original studies were included if they met each one of the following inclusion criteria:

1. developed *prognostic* models of hospital-induced delirium (a prognostic model is a formal combination of multiple variables which are used to predict whether an outcome will occur in an individual patient):

    a. primary outcome was the occurrence of hospital-induced delirium (i.e., delirium present: yes or no):

        i. the word "delirium" had to be used for the outcome instead of any similar term, such as "altered mental status", "confusion", or "neurological complication",

        ii. delirium was hospital-induced, i.e., absent on admission;

    b. models were developed using data from non-critically ill patients who were a minimum of 18 years old (we included adults of all ages because risk factors of hospital-induced delirium in older age may be present during the course of a lifespan) and hospitalized in non-intensive care medical, medical-surgical, or surgical units; if it was unclear where patients were hospitalized, we:

        i. assessed how the outcome of hospital-induced delirium was measured and only included studies which used delirium assessment tools for medical-surgical patients, such as the Confusion Assessment Method (for studies after 2001 when the Confusion Assessment Method for the Intensive Care Unit was developed), [19, 20]

        ii. excluded studies where patients were likely to be hospitalized in intensive care units following surgery, such as coronary artery bypass surgery and other serious surgeries;

2. validated their models using any one of the following three ways:

    a. internally by comparing the Akaike Information Criterion (AIC), Bayesian Information Criterion (BIC), Deviance Information Criterion (DIC), Mallow's Cp, or adjusted R-squared among various models which were developed using the same set of data (the various models had to be presented in the article or supplementary material),

b. internally by bootstrapping, cross-validating, or randomly splitting the population into training and test sets and developing the models using the training set and validating the models using the test set,

c. externally by comparing model performance between an internal dataset (dataset used to develop/train a model) and external dataset (data used to validate/test the model, for example, in a different population), where both the internal and external datasets came from the same study design.

Studies were excluded if they met any one of the following exclusion criteria:

1. did not have delirium as the primary outcome:

   a. had delirium as a predictor, for example, in a prediction model estimating the incidence of postoperative complications,

   b. predicted the course of delirium (for example, delirium severity) or outcomes of delirium (for example, post-delirium complications, delirium recovery, delirium survival, etc.),

2. developed *diagnostic* models (for example, studies assessing the accuracy of delirium assessment tools or studies validating delirium assessment tools),

3. failed to validate their prediction models using any one of the three ways which are listed in the inclusion criteria,

4. based in populations or settings other than inpatient medical and/or surgical units:

   a. community, including assisted living,

   b. emergency departments/rooms,

   c. gynecologic and/or obstetrical units,

   d. intensive care units,

   e. nursing homes/long-term care facilities,

   f. outpatient rehabilitation facilities,

   g. psychiatric hospitals/units,

   h. step-down units,

   i. total hospital patient population (because it was unclear what unit types were included);

5. lacked abstracts for the title and abstract screening or full texts for the full-text screening (including through the interlibrary loan system which is offered by our university).

## Selection process

The selection process consisted of two parts. The first part involved screening of the records which had been identified in the database search by title and abstract against the eligibility criteria. If the form of model validation was unclear in the abstract, the article was included and checked for appropriate validation in the full text. Reviews which seemed relevant were included and individual records were extracted and screened. Five percent of the records were independently screened by two researchers and the percentage of agreement was calculated. Any discrepancies were discussed with an objective to reach an agreement. For unresolved

discrepancies, the primary investigator was consulted. Once the final agreement had been reached, the remainder of unscreened records was halved, and each researcher independently reviewed one half.

The second part of the selection process involved full-text screening. The full texts of all the records which had been included in the title and abstract screening were independently screened against the eligibility criteria by two researchers. The percentage of agreement was calculated for the first 5% of the full texts. Any discrepancies were discussed with an objective to reach anagreement. For unresolved discrepancies, the primary investigator was consulted. Once the final agreement had been reached, the remainder of unscreened full texts was halved, and each researcher independently reviewed one half. The articles from each researcher were then added for inclusion in the final qualitative synthesis.

## Data extraction process and synthesis

The Checklist for Critical Appraisal and Data Extraction for Systematic Reviews of Prediction Modelling Studies (CHARMS) was used for data extraction [21]. This checklist provides a list of data items to be extracted from prediction model development and/or validation studies. Two researchers independently extracted data from each study. The extracted data items spanned the following domains: the source(s) of data, participants, outcome(s) to be predicted, candidate predictors, sample size, missing data, model development, model performance, model evaluation, results, and interpretation and discussion.

Meta-analysis was considered inappropriate for the purpose of our review; the purpose of our review was not to assess any specific associations between the predictors and outcome, but to assess risk of bias in existing prognostic models of hospital-induced delirium for medical-surgical units. Instead, a qualitative synthesis was conducted. All of the studies were included in the qualitative synthesis.

Specific data items were summarized to address the purpose of our review. The data items included the first author(s) and year, design, data source(s), study dates, inclusion criteria, measure of delirium, sample size, number of subjects with delirium, number of subjects without delirium, statistical model(s), sensitivity and specificity, area under the receiver operating curve (AUROC), negative and positive predictive values, and type(s) of validation method(s). For model development studies with prospective validation cohorts, the sample sizes and numbers of subjects with delirium from the development and validation cohorts were added to present the total sample sizes and numbers of subjects with delirium. When an article did not report negative and positive predictive values but did report a confusion matrix, the negative and positive predictive values were calculated.

The data items were tabulated to facilitate the identification of patterns in the data. The studies were chronologically ordered by date of publication. Ranges of the data items in each column were determined to summarize the data in the table. In addition, median sample size and median number of subjects with delirium were calculated. Except for the AUROCs, average sensitivity, specificity, and negative and positive predictive values were not calculated because most of these data items were not reported in the articles. We did not contact the study authors to provide the unavailable data.

## Applicability

The Prediction Model Risk of Bias Assessment Tool (PROBAST) was used to assess applicability of each study to the systematic review purpose of assessing risk of bias in existing prognostic models of hospital-induced delirium for medical-surgical units [22]. Applicability was assessed across three domains: participants, predictors, and outcome. Concerns about the

applicability of a study to the review purpose may arise when the participants, predictors, or outcome of the study differ from those specified in the review purpose [22].

One researcher assessed each study for applicability. The researcher rated each domain as low, high, or unclear concern for applicability based on the information reported in the article and its supplementary material. The researcher then rated overall applicability. All of the domains had to be rated as "low concern" for the study to have a low concern about applicability. If any domain was rated as "high concern", the study had a high concern about applicability. A study had an unclear concern about applicability if any number of domains was rated as "unclear concern" as long as all of the remaining domains were rated as "low concern".

### Reporting bias assessment

The CHARMS was used to assess the reporting bias. The checklist consists of 35 items. One item (item #4: "Details of treatments received, if relevant") was omitted from the assessment because none of the included studies was of experimental design. Each study was able to score the maximum number of 34 points, unless any other item(s) was also inapplicable to the study. For example, model validation studies were not scored on items #24, 25, 26, 31, and 32. Two researchers independently completed the CHARMS for each study. Every time a checklist item was available for extraction from a study, the item was marked as present and assigned a "1". Otherwise, a "0" was entered. The percentage of agreement was calculated to measure the reliability of the rating process. Any discrepancies were discussed and resolved by an agreement between the two researchers.

### Risk of bias assessment

The PROBAST was used to assess risk of bias [22]. Bias is defined as presence of systematic error which leads to distorted results, limiting internal validity of a study. PROBAST is specifically applicable for use in systematic reviews of prediction model development and/or validation studies. The 20 signaling questions address sources of bias across four domains: participants, predictors, outcome, and analysis. Presence of bias in any of these domains can influence the predictive performance of prediction models.

Two researchers, one of them being a statistical expert, independently assessed each study for risk of bias. The researchers rated each signaling question as "Yes", "No", or "No information" based on the information reported in the article and its supplementary material. The percentage of agreement was calculated to measure the reliability of the rating process. Any discrepancies were discussed and resolved by an agreement between the two researchers.

Risk of bias was then rated across four domains: participants, predictors, outcome, and analysis. All of the signaling questions in a domain had to be rated as "Yes" for the domain to be at low risk of bias. If any signaling question was rated as "No", a domain was at high risk of bias. A domain had unclear risk of bias if any number of signaling questions was rated as "No information" as long as all of the remaining signaling questions were rated as "Yes". In addition to these rules, the researcher was also able to exercise judgement in determining risk of bias for each domain. For example, any domain could still be considered to be at low risk of bias despite having all of the signaling questions rated as "No".

## Results

### Study selection

Fig 1 presents the PRISMA diagram. The database search yielded 5,650 records: 3,453 from Web of Science Core Collection, 1,971 from PubMed, 214 from CINAHL, and 12 from APA PsycInfo. There were 1,309 duplicates. After the duplicates had been removed, 4,341 unique

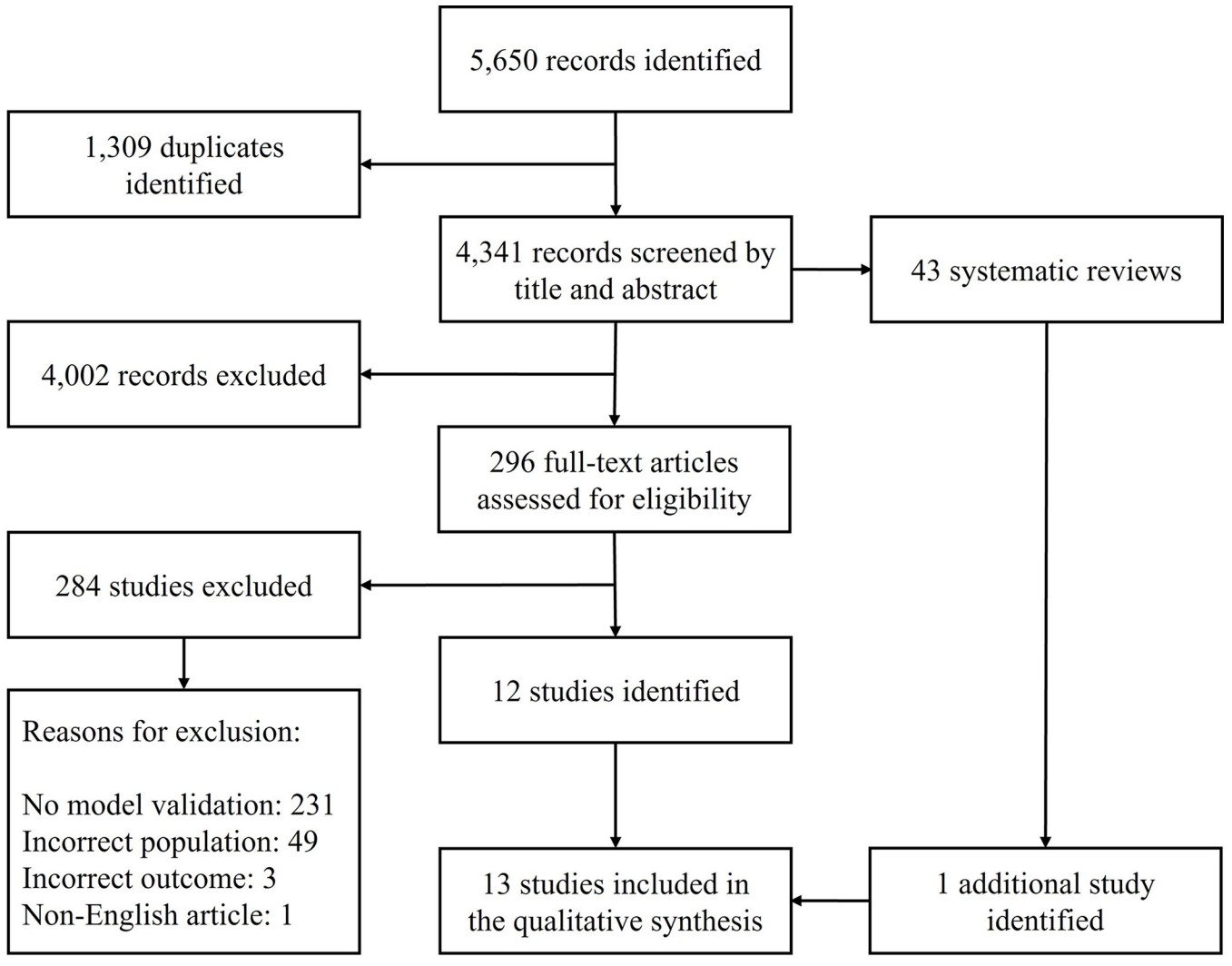

**Fig 1. PRISMA diagram.** PRISMA = Preferred Reporting Items for Systematic Reviews and Meta-Analyses.

records were left for the title and abstract screening which resulted in the exclusion of 4,002 records. The percentage of agreement for this step was 79%. All of the discrepancies were discussed and resolved through agreement.

There were 339 articles remaining, including 43 systematic reviews. The 296 articles were screened by full text. The full-text screening resulted in the exclusion of 284 additional studies. The percentage of agreement for this step was 85%. All of the discrepancies were discussed and resolved through agreement.

A total of 231 studies were excluded due to the lack of appropriate model validation as specified in our inclusion criteria. Forty-nine studies were excluded because they included patients from settings other than inpatient medical and/or surgical units, mainly intensive care units. Of note, we excluded studies which included mixed patient populations from both intensive care and medical-surgical units. Three studies were excluded because the outcome was different than hospital-induced delirium; an example was delirium 30 days after discharge. One study was excluded because the full text of the article was unavailable in English after a request had been made via our university interlibrary loan system.

Twelve studies remained for inclusion in the qualitative synthesis. The review of the systematic reviews resulted in the identification of an additional study. Finally, 13 studies were included in the qualitative synthesis [8, 23–34]. Out of the 13 studies, there were 10 model development and validation studies [8, 23–25, 27–31, 33] and 3 model validation only studies [26, 32, 34].

## Study characteristics

Table 1 presents characteristics of the included studies. Prognostic models of hospital-induced delirium among adult medical-surgical patients were developed across a variety of countries, including the United States (n = 5), [8, 23, 24, 27, 30] the Netherlands (n = 3), [26, 31, 34] Austria (n = 1), [32] Chile (n = 1), [29] China (n = 1), [33] Japan (n = 1), [28] and the United Kingdom (n = 1) [25] between 1993 and 2022. Most of the studies were prospective cohorts (n = 10). [8, 23–27, 29, 30, 32, 33] Two studies were retrospective cohorts, [28, 31] and one was a secondary analysis of a prospective cohort study [34].

Seven studies developed and/or validated prognostic models of hospital-induced delirium in the medical patient population. [8, 24, 25, 28, 29, 31, 33] Four studies developed and/or validated prognostic models of postoperative delirium in the surgical patient population [26, 27, 30, 34]. Two studies used a mixed medical and surgical patient population [23, 32].

Hospital-induced delirium was measured differently across the studies. The Confusion Assessment Method was the most common measure [8, 23, 24, 26, 27, 29, 30, 33]. Other measures included the Diagnostic Statistical Manual of Mental Disorders, [23, 25, 26, 28, 34]. Delirium Observation Screening Scale, [31, 34] delirium diagnoses,[31, 32] and standardized chart review method for delirium.[30]

The 13 studies included 14,317 subjects. A total of 1,049 developed hospital-induced delirium. The sample sizes ranged from 184 to 5,530 subjects with a median of 566. The number of subjects with hospital-induced delirium ranged from 25 to 150 with a median of 74.

## Model performance

Thirteen unique prognostic models of hospital-induced delirium among adult medical-surgical patients were developed across the 10 model development and validation studies (some studies developed multiple models). Table 2 presents information about the models. Traditional statistical methods, i.e., methods which did not involve machine learning, were used for 11 models [8, 23–25, 27, 29–31]. Machine learning was used for 2 models [28, 33]. There were 8 models with reported AUROCs across the model development and validation studies [8, 23, 25, 28–31, 33]. The AUROCs after validation ranged from 0.64 to 0.95. The average AUROC was 0.76. Li et al.'s (2022) model has the highest AUROC (0.95) [33].

Half of the model development and validation studies attempted to internally validate their models [27, 28, 30, 31, 33]. One method of internal validation was generally employed. Two studies split their samples into development and validation sets [28, 33]. Bootstrapping was used in two studies [27, 30] and cross-validation was used in one study [31]. Another half of the model development and validation studies attempted to externally validate their models in the form of prospective validation [8, 23–25, 29]. All of the model validation studies were independent validation studies, i.e., studies which were conducted by investigators who were independent from those who developed the original models [26, 32, 34].

## Applicability of studies

Table 3 presents overall ratings of applicability and by domain for each study. All of the studies were rated as "low concern" for the participants, predictors, and outcome domains. All of the studies were consequently rated as "low concern" overall.

**Table 1. Study characteristics.**

| Author Year | Design | Data Source | Study Dates | Inclusion Criteria | Measure of Delirium | Sample Size[a] | Delirium[a] | No Delirium[a] |
|---|---|---|---|---|---|---|---|---|
| **Inouye 1993** [8] | Prospective cohort | Yale-New Haven Hospital, CT | 06/1988–06/1990 | Pts ≥70 yrs admitted to general medicine floor for ≥48 hrs | CAM | 281 | 56 | 225 |
| **Pompei 1994** [23] | Prospective cohort | University of Chicago Hospitals, IL Yale-New Haven Hospital, CT | 11/1989–06/1991 | Pts ≥65 yrs (derivation set) or ≥70 yrs (test set) admitted to medical/surgical ward for ≥48 hrs | CAM, DSM-III-R | 755 | 150 | 605 |
| **Inouye & Charpentier 1996** [24] | Prospective cohort | Yale-New Haven Hospital, CT | 11/06/1989–07/31/1991 | Pts ≥70 yrs admitted to general medicine floor for ≥48 hrs | CAM | 508 | 82 | 426 |
| **O'Keeffe & Lavan 1996** [25] | Prospective cohort | Royal Liverpool University Hospital, the UK | - | Pts admitted to acute-care geriatric unit w/ anticipated stay of ≥48 hrs | DSM-III | 184 | 53 | 131 |
| **Kalisvaart 2006** [26] | Prospective cohort | Medical Center Alkmaar, the Netherlands | 08/2000–08/2002 | Pts ≥70 yrs undergoing hip surgery | CAM, DSM-IV | 603 | 74 | 529 |
| **Leung 2007** [27] | Prospective cohort | University of California San Francisco Medical Center, CA | 2001–2006 | Pts ≥65 yrs undergoing major elective non-cardiac surgery w/ anticipated stay of >48 hrs | CAM | 190 | 29 | 161 |
| **Kobayashi 2013** [28] | Retrospective cohort | St. Luke's International Hospital, Japan | 04/01/2009–03/31/2010 | Adult pts admitted to internal medicine unit | DSM-IV | 3,570 | 142 | 3,428 |
| **Carrasco 2014** [29] | Prospective cohort | University hospital affiliated w/ the Pontifical Catholic University of Chile | - | Pts ≥65 yrs admitted to general medical ward in the previous 48 hrs | CAM | 478 | 37 | 441 |
| **Jones 2016** [30] | Prospective cohort | Beth Israel Deaconess Medical Center, MA Brigham and Women's Hospital, MA | 06/18/2010–08/08/2013 | Pts ≥70 yrs undergoing major elective non-cardiac surgery w/ anticipated stay of ≥3 days | CAM, standardized chart review method for delirium | 566 | 135 | 431 |
| **Neefjes 2017** [31] | Retrospective cohort | VUmc Cancer Center Amsterdam, the Netherlands | 01/01/2011–09/2013 | Pts w/ solid malignancies admitted to medical oncology ward | Diagnosis of delirium in EHR, DOSS | 620 | 98 | 522 |
| **Jauk 2020** [32] | Prospective cohort | Steiermärkische Krankenan-staltengesellschaft m.b.H. (KAGes), Austria | 06/01/2018–12/31/2018 | Pts ≥18 yrs admitted to internal medicine or surgical department | ICD codes | 5,530 | 67[b] | 5,449 |
| **Li 2022** [33] | Prospective cohort | West China Hospital of Sichuan University, China | 03/2016–01/2017 | Pts ≥70 yrs admitted to internal medicine ward | CAM | 740 | 101 | 639 |
| **Wong 2022** [34] | Secondary analysis of prospective cohort | University Medical Center Groningen, the Netherlands | 10/01/2011–06/01/2012 | Pts ≥50 yrs undergoing various surgical procedures | DOSS, DSM-IV-TR | 292 | 25 | 267 |

*Note.* CAM = Confusion Assessment Method; DOSS = Delirium Observation Screening Scale; DSM = Diagnostic and Statistical Manual of Mental Disorders;

EHR = electronic health record; ICD = International Classification of Diseases; pts = patients; yrs = years.

[a] Total number from both cohorts for studies with the development and prospective validation cohorts.

[b] Only includes the delirium cases which were coded using the ICD-10 code "F05" (delirium due to known physiological condition). We did not include the "F10.4"

model in our qualitative synthesis because this model is for delirium due to alcohol withdrawal and not hospital-induced delirium which is the focus of our systematic

review.

## Reporting bias

The percentage of agreement on the CHARMS between the two researchers was 90%. All of the discrepancies were discussed and resolved through agreement. Each study was rated across the maximum of 34 items. Additional items (up to 6) were irrelevant for 6 studies. Therefore,

reporting of the CHARMS items was calculated in percentages (Fig 2). The highest reporting of the CHARMS items was 88%.[33] Two other studies had at least 80%.[24, 34] The lowest reporting of the CHARMS items was 61%.[31]

Because certain CHARMS items were inapplicable to some studies, the percentage of studies rather than the number of studies which reported on individual CHARMS items is reported. Some CHARMS items were more commonly reported than others (Fig 3). In fact, more than half of the CHARMS items were reported by at least 75% of the studies. Fifteen items (3, 6, 7, 8, 10, 12, 17, 18, 22, 24, 25, 29, 31, 34, and 35) were reported in 100% of the studies. The CHARMS items with the least reporting were items 9 (23%), 15 (23%), 23 (23%), 26 (0%), 27 (15%), and 30 (8%).

## Risk of bias in studies

The percentage of agreement on the PROBAST between the two researchers was 79%. All of the discrepancies were discussed and resolved through agreement. Each study was first rated

**Table 2. Model performance.**

| Author and Year | Statistical Model (Name of the Model, if Applicable) | Sensitivity/ Specificity | AUROC | NPV/PPV | Validation Method |
|---|---|---|---|---|---|
| **Inouye 1993 [8]** | Proportional hazards model → Risk stratification model | Multiple sensitivities reported for different risk strata | Development cohort: 0.74 Validation cohort: 0.66 | - | Development and prospective validation cohorts |
| **Pompei 1994 [23]** | Logistic regression → Risk stratification model | Multiple sensitivities reported for different risk strata | Development set: 0.74 Test set: 0.64 | - | Derivation and prospective test set |
| **Inouye & Charpentier 1996 [24]** | Binomial regression → Risk stratification model | Multiple sensitivities reported for different risk strata | - | - | Development and prospective validation cohorts |
| **O'Keeffe & Lavan 1996 [25]** | Logistic regression → Risk stratification model | Multiple sensitivities reported for different risk strata | Development group: 0.79 Validation group: 0.75 | - | Derivation and prospective validation groups |
| **Kalisvaart 2006 [26]** | Inouye et al.'s (1993) model (proportional hazards model) | Multiple sensitivities reported for different risk strata | 0.73 | - | External validation cohort |
| **Leung 2007 [27]** | Logistic regression | - | - | - | Bootstrapping |
| **Kobayashi 2013 [28]** | Decision tree (Chi-Square Automatic Interaction Detector algorithm) | - | Development group: 0.82 Validation group: 0.82 | - | Random split into development and validation groups |
| | Logistic regression | - | Development group: 0.78 Validation group: 0.79 | - | |
| **Carrasco 2014 [29]** | Logistic regression → Linear prediction rule ("delirium predictive risk score") | 0.88/0.74 for the cut-off point of -240 | Development cohort: 0.86 Validation cohort: 0.78 | - | Development and prospective validation cohorts |
| **Jones 2016 [30]** | Logistic regression (the "bivariable model") | - | - | - | Bootstrapping |
| | Logistic regression (the "multivariable model I") | - | - | - | |
| | Logistic regression (the "multivariable model II") | - | - | - | |
| **Neefjes 2017 [31]** | Tree analysis | Before validation:—After validation: 0.4/0.85 | Before validation: 0.81 After validation: 0.65 | - | Cross-validation |
| **Jauk 2020 [32]** | Kramer et al.'s (2017) [35] model (random forest) | 0.741/0.822 | 0.855 | 0.995/0.058 | External validation cohort |
| **Li 2022 [33]** | Decision tree (Classification and Regression Trees algorithm) | Training set:—Test set: 0.933/943 | Training set: 0.967 Test set: 0.950 | Training set:—Test set: 0.989/0.718 | Training and test sets |

(*Continued*)

**Table 2.** (Continued)

| Author and Year | Statistical Model (Name of the Model, if Applicable) | Sensitivity/ Specificity | AUROC | NPV/PPV | Validation Method |
|---|---|---|---|---|---|
| Wong 2022 [34] | Carrasco et al.'s (2014) [29] model (logistic regression) | 0.565/0.637 | 0.563 | - | External validation cohort |
| | Dai et al.'s (2000) [36] model (logistic regression) | 0.920/0.508 | 0.739 | - | |
| | de Wit et al.'s (2016) [37] model (logistic regression) | 0.600/0.715 | 0.635 | - | |
| | Ettema et al.'s (2018) [38] model (logistic regression) | 0.417/0.759 | 0.580 | - | |
| | Freter et al.'s (2005) [39] model (logistic regression) | 0.560/0.591 | 0.576 | - | |
| | Halladay et al.'s (2018) [40] model (random forest) | 0.300/0.739 | 0.519 | - | |
| | Kim et al.'s (2016) [41] model (logistic regression) | 0.300/0.919 | 0.610 | - | |
| | Litaker et al.'s (2001) [42] model (logistic regression) | 0.520/0.860 | 0.706 | - | |
| | Pendlebury et al.'s (2017) [43] model (logistic regression) | 0.400/0.678 | 0.539 | - | |
| | Pompei et al.'s (1994) [23] model (logistic regression) | 0.520/0.582 | 0.543 | - | |
| | Rudolph et al.'s (2009) [44] model (logistic regression) | 0.833/0.356 | 0.610 | - | |
| | Rudolph et al.'s (2016) [45] model (logistic regression) | 0.750/0.442 | 0.624 | - | |
| | ten Broeke et al.'s (2018) [46] model (logistic regression) | 0.680/0.574 | 0.635 | - | |
| | Zhang et al.'s (2019) [47] model (logistic regression) | 0.760/0.553 | 0.650 | - | |

*Note.* AUROC = area under the receiver operating curve; NPV = negative predictive value; PPV = positive predictive value.

across 20 signaling questions. Three model validation studies were rated across 17 signaling questions, because three signaling questions (4.5, 4.8, and 4.9) in the "analysis" domain were only applicable to model development studies. Hence, the percentage of studies instead of the number of studies is reported. Fig 4 presents the percentage of studies with the signaling questions rated as "Yes", "No", or "No Information". Table 3 presents overall risk of bias and by domain for each study.

In the "participants" domain, the risk of bias was high in 31% of the studies, unclear in 8% of the studies, and low in 61% of the studies. The source of high bias in this domain was the signaling question 1.2 ("Were all inclusions and exclusions of participants appropriate?") which was rated as "No" in 38% of the studies. These studies typically failed to report whether they had excluded prevalent cases of delirium.

The "predictors" domain had 77% of the studies with a low risk of bias. This was the only domain with all of the signaling questions rated as "Yes" in at least 75% of the studies. The signaling question which was rated as "Yes" in all of the studies was 2.3 ("Are all predictors available at the time the model is intended to be used?"). Only 15% of the studies were at a high risk of bias in this domain.

The "outcome" domain had the most studies with an unclear risk of bias (54%). The source of this was the signaling question 3.5 ("Was the outcome determined without knowledge of

**Table 3. PROBAST results\*: Domain and overall applicability and ROB by study.**

| Study | Applicability | | | ROB | | | | Overall | |
|---|---|---|---|---|---|---|---|---|---|
| | Participants | Predictors | Outcome | Participants | Predictors | Outcome | Analysis | Applicability | ROB |
| Inouye 1993 [8] | + | + | + | + | − | − | − | + | − |
| Pompei 1994 [23] | + | + | + | + | − | − | − | + | − |
| Inouye & Charpentier 1996 [24] | + | + | + | + | + | ? | − | + | − |
| O'Keeffe & Lavan 1996 [25] | + | + | + | ? | + | ? | − | + | − |
| Kalisvaart 2006 [26] | + | + | + | + | + | ? | − | + | − |
| Leung 2007 [27] | + | + | + | − | ? | ? | − | + | − |
| Kobayashi 2013 [28] | + | + | + | − | + | ? | − | + | − |
| Carrasco 2014 [29] | + | + | + | + | + | ? | − | + | − |
| Jones 2016 [30] | + | + | + | + | + | − | − | + | − |
| Neefjes 2017 [31] | + | + | + | − | + | − | − | + | − |
| Jauk 2020 [32] | + | + | + | − | + | − | − | + | − |
| Li 2022 [33] | + | + | + | + | + | + | − | + | − |
| Wong 2022 [34] | + | + | + | + | + | ? | − | + | − |

*Note*. PROBAST = Prediction Model Risk of Bias Assessment Tool; ROB = risk of bias.

\* "+" indicates low ROB/low concern about applicability; "−" indicates high ROB/high concern about applicability; and "?" indicates unclear ROB/unclear concern about applicability.

predictor information?") which was rated as "No information" in 77% of the studies. Blinding of the outcome to the predictors was reported by only 23% of the studies. The source of high bias in this domain was the signaling question 3.4 ("Was the outcome defined and determined in a similar way for all participants?") which was rated as "No" in 38% of the studies. Only 8% of the studies were at low risk of bias in this domain.

The "analysis" domain was at high risk of bias across all of the studies. The signaling question which was always rated as "No" was question 4.1 ("Were there a reasonable number of participants with the outcome?"). Other signaling questions which were rated as "No" in more than half of the studies were 4.8 ("Were model overfitting and optimism in model performance accounted for?"), 4.5 ("Was selection of predictors based on univariable analysis avoided?"), 4.4 ("Were participants with missing data handled appropriately?"), and 4.7 ("Were relevant model performance measures evaluated appropriately?").

Finally, overall risk of bias was determined based on the domain ratings. Overall risk of bias was rated as high when at least one of the four domains was rated as high. All of the studies had at least one domain which was at high risk of bias. Consequently, all of the studies had a high overall risk of bias (Table 3). Six studies had a high risk of bias in the "analysis" domain only.[24–26, 29, 33, 34] Three studies had a high risk of bias across two domains.[27, 28, 30] Four studies had a high risk of bias across three domains.[8, 23, 31, 32] No study had a high risk of bias across all of the four domains.

## Discussion

This systematic review assessed risk of bias in existing prognostic models of hospital-induced delirium for medical-surgical units. We identified two challenges which may limit the validity of existing prognostic models of hospital-induced delirium for medical-surgical units. First, there was a high risk of bias in each study. Second, external validity of models was tested at low levels of transportability.

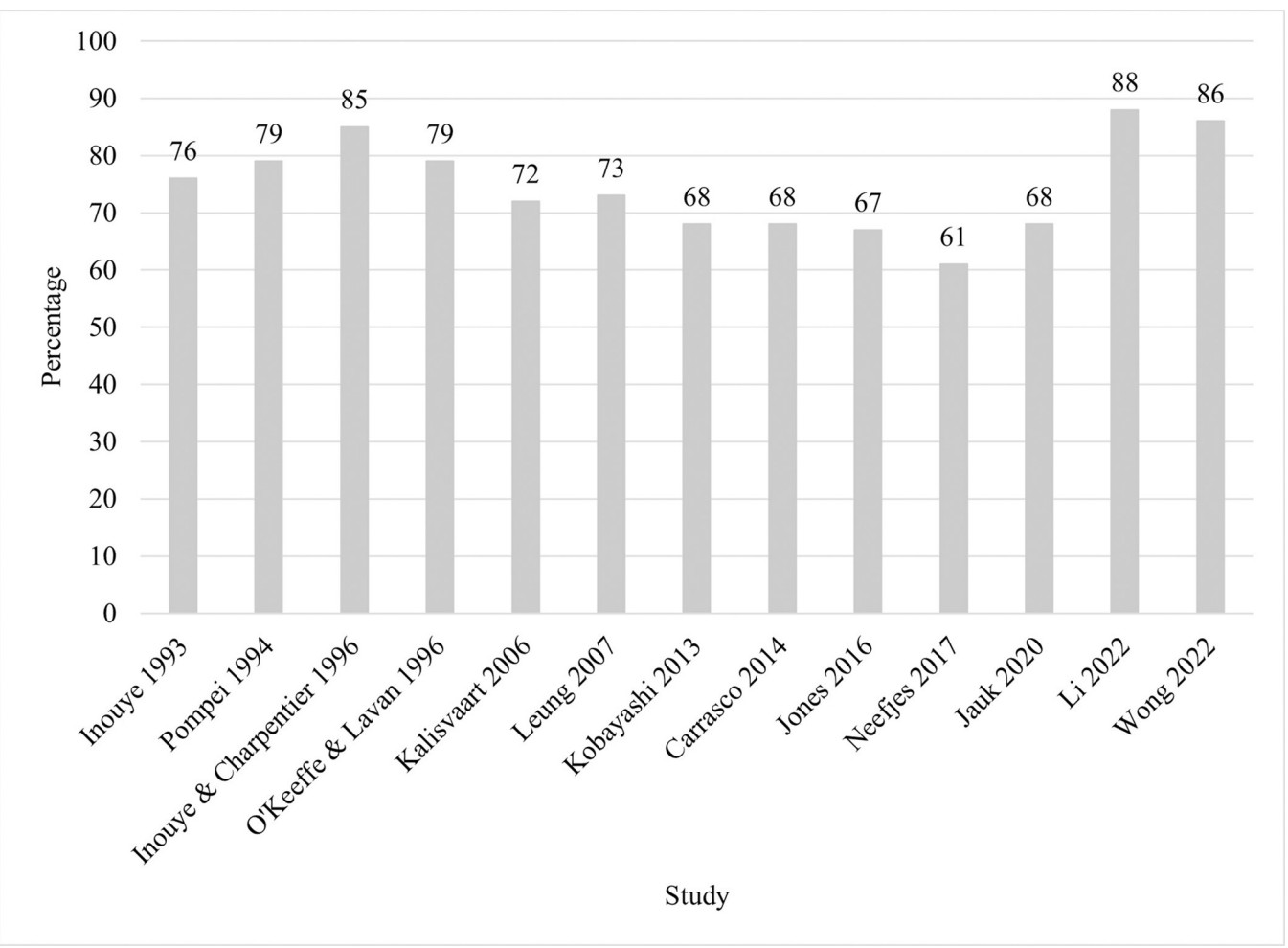

**Fig 2. CHARMS Results*: Reporting of the CHARMS items.** CHARMS = Checklist for Critical Appraisal and Data Extraction for Systematic Reviews of Prediction Modelling Studies. * The studies are chronologically ordered by date of publication.

## Challenge #1: High risk of bias in model development and validation

The statistical analysis was the greatest source of bias. The top three problems were failure to include a sufficient number of participants with the outcome (PROBAST signaling question 4.1), failure to address optimism in model performance (PROBAST signaling question 4.8), and failure to avoid the selection of predictors based on univariable analysis (PROBAST signaling question 4.5)

**4.1: Were there a reasonable number of participants with the outcome?.** No model development and validation study had a sufficient sample size in the development cohort to allow for adequate numbers of participants with the outcome in relation to the numbers of candidate predictors. The number of events, i.e., the smaller number between the number of participants with the outcome and the number of participants without the outcome, [48] needed to be greater than 20 to minimize overfitting.[22] Similarly, no model validation study had a sufficient sample size to allow for adequate numbers of participants with the outcome. The number of participants with the outcome needed to be at least 100 to minimize overfitting [22].

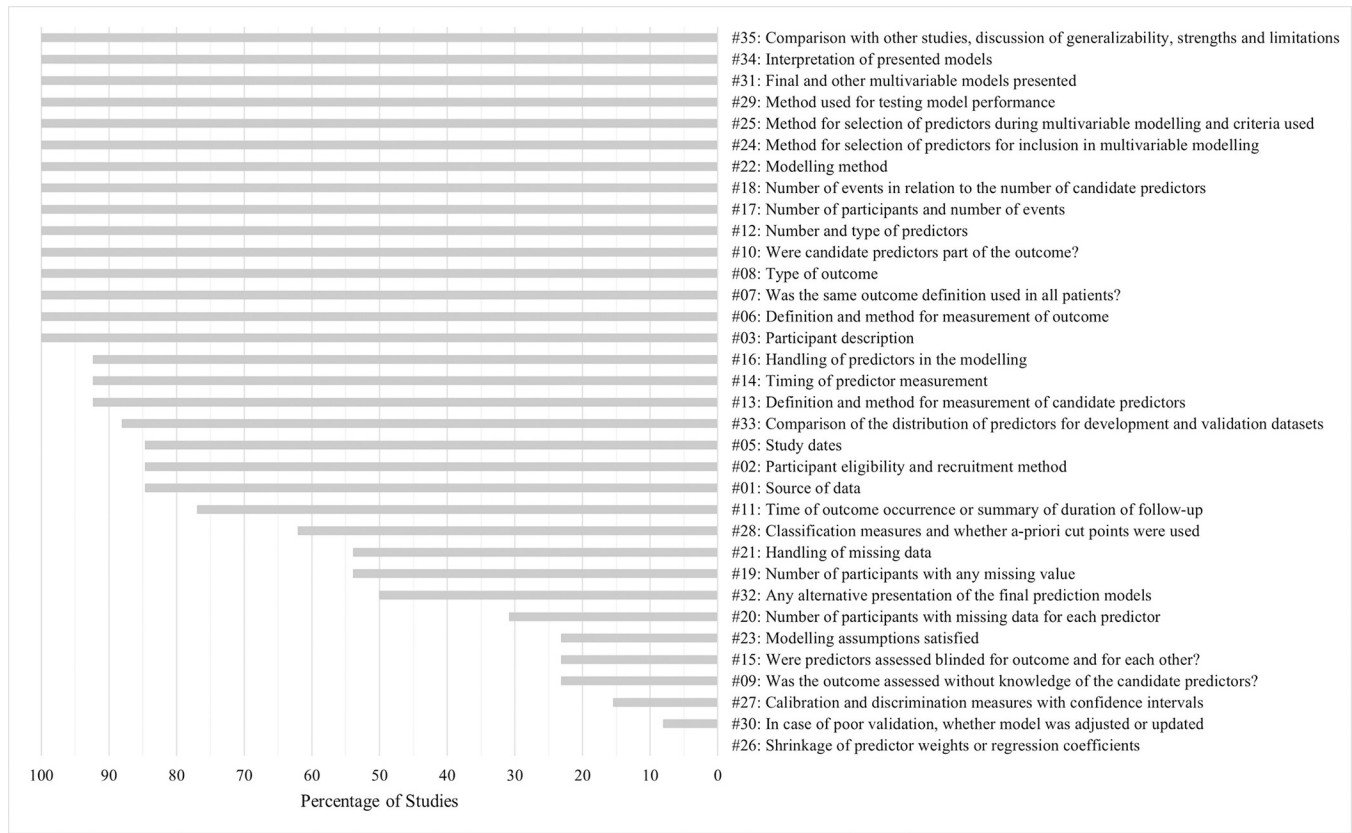

**Fig 3. CHARMS Results: Percentage of studies reporting on individual CHARMS items.** CHARMS = Checklist for Critical Appraisal and Data Extraction for Systematic Reviews of Prediction Modelling Studies.

**4.8: Were model overfitting and optimism in model performance accounted for?.** Half of model development and validation studies in this systematic review performed internal validation. Validation is important to adjust for optimism of a model by evaluating or testing the performance of the model. The simplest technique of internal validation is to randomly split the data into two parts, one to train and another to test the model [49]. Two studies used the split-sample technique [28, 33]. However, this technique is discouraged because it tends to underestimate the performance of a model [50]. A more accurate and sophisticated technique is cross-validation [50]. On the other hand, the bootstrap resampling technique is the most accurate in estimating the true performance of a model [50]. Although three model development and validation studies used either cross-validation or bootstrapping, [27, 30, 31] none used these techniques appropriately by including all steps of model development in the internal validation process [22].

**4.5: Was selection of predictors based on univariable analysis avoided?.** Most model development and validation studies relied on univariable analyses for selection of the candidate predictors to include in the multivariable modelling [8, 23–25, 27–29]. This may have inadvertently excluded important predictors which were only significant in the context of other predictors (for example, through interaction) or included unimportant candidate predictors which just happened to be associated with the outcome due to a confounding effect [22]. Candidate predictors should be included in multivariate modelling on the basis of existing knowledge regardless of statistical significance [22]. Alternatively, selection of candidate predictors for inclusion in multivariate modelling can be supported with statistical methods

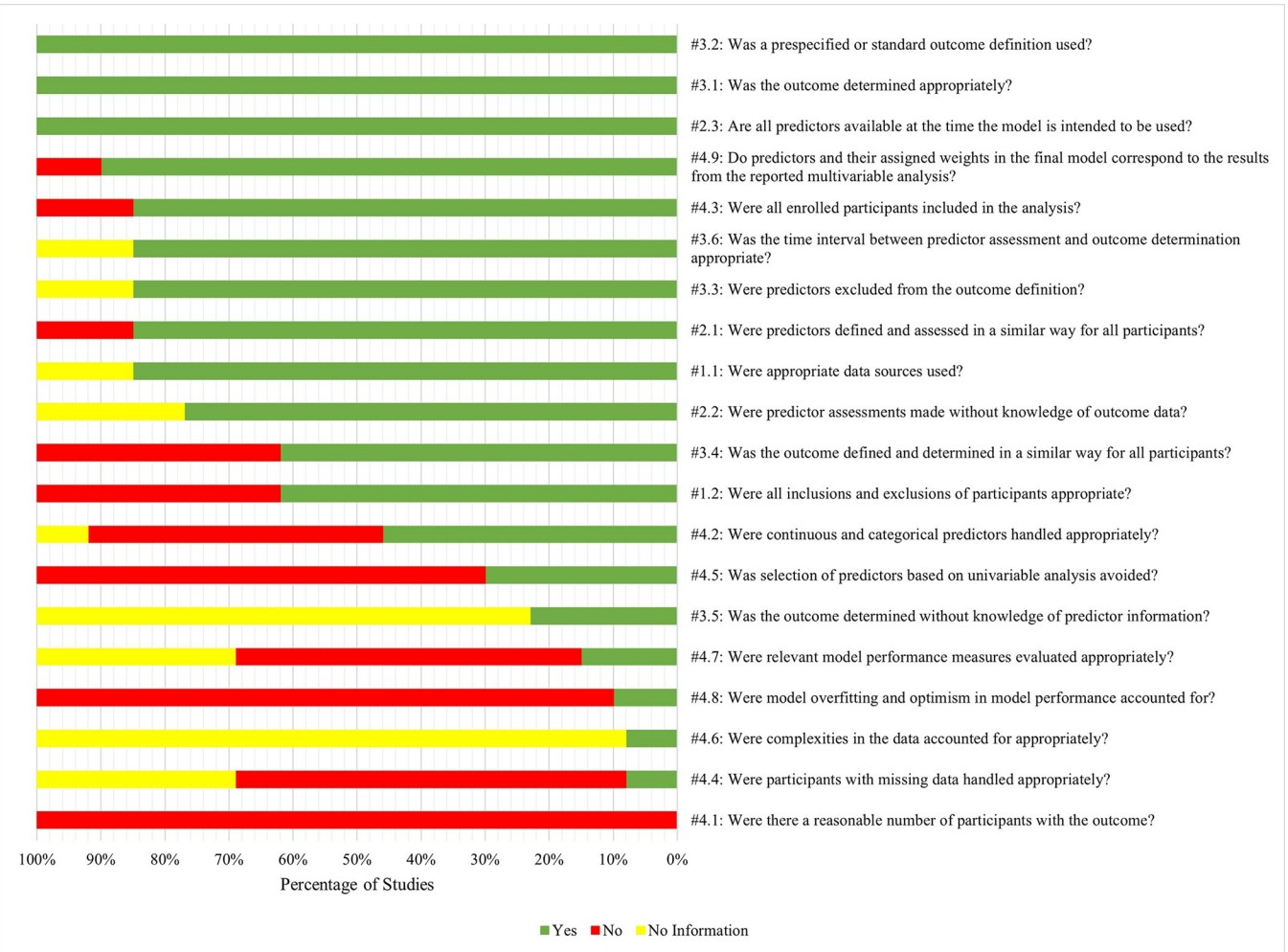

**Fig 4. PROBAST results: Percentage of studies rated as Y, N, or NI by signaling question.** N = No; NI = No information; PROBAST = Prediction Model Risk of Bias Assessment Tool; Y = Yes.

which are not based on tests between individual predictors and the outcome, such as the principal component analysis [22].

## Challenge #2: Low level of external validation

Transportability was tested in 5 models across the 10 model development and validation studies and 16 models across the 3 model validation studies. The external validity of a model is established by replicating its accuracy across levels of external validation which represent cumulative types of transportability of a model: prospective validation (level 1), independent validation (level 2), multisite validation (level 3), multiple independent validations (level 4), and multiple independent validations with varying follow-up periods (level 5) [51]. No model of hospital-induced delirium for medical-surgical units was tested at the fifth, fourth, or third level of external validation in this systematic review. The highest level of external validation was level 2. Sixteen models were tested at the second level [26, 32, 34]. Five models were tested at the first level [8, 23–25, 29].

## Limitations

Our systematic review has some limitations. Because we focused on prognostic models of hospital-induced delirium for medical-surgical units, our systematic review is limited by the general conceptualization of medical-surgical units as units which admit patients of the lowest level of acuity. However, patients who are hospitalized in medical-surgical units may vary in acuity of care.

We decided to only include patients from medical-surgical units to ensure a homogenous sample of studies because we recognize that inpatient populations may differ with regards to risk factors of hospital-induced delirium. Only applicable studies were included based on the inclusion criteria which specifically described our outcome and patient population of interest. We may have inadvertently omitted studies which included medical-surgical patients by excluding articles where the units were unclear.

Our review was limited to the English language studies. Therefore, valid prognostic models of hospital-induced delirium at low overall risk of bias may exist but have been reported in non-English language literature. In fact, there was one study which we had to exclude because it was written in a language other than English.

## Conclusion

Our findings highlight the ongoing scientific challenge of developing a valid prognostic model of hospital-induced delirium for medical-surgical units to tailor preventive interventions to patients who are at high risk of this iatrogenic condition. With limited knowledge about generalizable prognosis of hospital-induced delirium in medical-surgical units, existing prognostic models should be used with caution when creating clinical practice policies. Future research protocols must include robust study designs which take into account the perspectives of clinicians to identify and validate risk factors of hospital-induced delirium for accurate and generalizable prognosis in medical-surgical units.

## Supporting information

**S1 File.**
(RAR)

## Author Contributions

**Conceptualization:** Robert J. Lucero.

**Formal analysis:** Urszula A. Snigurska, Yiyang Liu, Sarah E. Ser, Tamara G. R. Macieira.

**Funding acquisition:** Ragnhildur I. Bjarnadottir, Robert J. Lucero.

**Methodology:** Urszula A. Snigurska, Margaret Ansell, David Lindberg, Robert J. Lucero.

**Writing – original draft:** Urszula A. Snigurska.

**Writing – review & editing:** Urszula A. Snigurska, Yiyang Liu, Sarah E. Ser, Tamara G. R. Macieira, Margaret Ansell, David Lindberg, Mattia Prosperi, Ragnhildur I. Bjarnadottir, Robert J. Lucero.

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
