## [Decision Letter · Decision Letter 0]

10 Apr 2023

PONE-D-22-28224Risk of bias in prognostic models of hospital-induced delirium for medical-surgical units: A systematic reviewPLOS ONE

Dear Dr. Snigurska,

Thank you for submitting your manuscript to PLOS ONE. After careful consideration, we feel that it has merit but does not fully meet PLOS ONE’s publication criteria as it currently stands. Therefore, we invite you to submit a revised version of the manuscript that addresses the points raised during the review process.

Kindly go through the reviewer's comment and submit a revised version with track changes.  

We look forward to receiving your revised manuscript.

Kind regards,

Blessing Onyinye Ukoha-kalu, Ph.D

Guest Editor

PLOS ONE

“This research was supported by the National Institute of Aging [R33 AG062884-04 to R. I. B. and R. J. L.].”

“This research was supported by the National Institute of Aging (https://www.nia.nih.gov/), R33 AG062884-04 to R. I. B. and R. J. L. The funders had no role in study design, data collection and analysis, decision to publish, or preparation of the manuscript.”

Reviewers' comments:

Reviewer's Responses to Questions

**Comments to the Author**

1. Is the manuscript technically sound, and do the data support the conclusions?

Reviewer #1: Yes

Reviewer #2: Yes

2. Has the statistical analysis been performed appropriately and rigorously? 

Reviewer #1: Yes

Reviewer #2: Yes

3. Have the authors made all data underlying the findings in their manuscript fully available?

Reviewer #1: Yes

Reviewer #2: Yes

4. Is the manuscript presented in an intelligible fashion and written in standard English?

Reviewer #1: Yes

Reviewer #2: Yes

5. Review Comments to the Author

Reviewer #1: The manuscripts was well written with just minor correction "Every year delirium complicates hospital stays for greater than 2.3 million adults of 65 years", The number of events, i.e., the smaller (should be the small) of the number of participants with the outcome and the number of participants without the outcome,[35].

I would like for the authors to elaborate on the consensus? The consensus technique employed, is it a Delphi technique? and the consensus reached by the two independent reviewer was two, a references to this should be stated to support that two independent reviewer can be used to reach a consensus.

Reviewer #2: The authors have carried out an interesting systematic review of hospital induced delirium in medical and surgical wards comprising patients managed by the medical and surgical units.

This study appropriately followed the standard process of a systematic review manuscript and the methodology and results were appropriately selected. The conclusion is in tandem with the evidence provided from the study.

ln page 9, General surgical hospitals have both medical and surgical patients wards, l am a little worried how those patients in the General Hospital wards that might have both medical and surgical patients were also excluded from the study?

ln page 15, understand the review of the articles were those written in English, what efforts were made to get appropriate English language translational application to get some of the excluded journal articles that were excluded in the study please?

6. PLOS authors have the option to publish the peer review history of their article (what does this mean?). If published, this will include your full peer review and any attached files.

Reviewer #1: **Yes: **Chigozie Gloria Anene-Okeke

Reviewer #2: No

While revising your submission, please upload your figure files to the Preflight Analysis and Conversion Engine (PACE) digital diagnostic tool, https://pacev2.apexcovantage.com/. PACE helps ensure that figures meet PLOS requirements. To use PACE, you must first register as a user. Registration is free. Then, login and navigate to the UPLOAD tab, where you will find detailed instructions on how to use the tool. If you encounter any issues or have any questions when using PACE, please email PLOS at figures@plos.org. Please note that Supporting Information files do not need this step.<quillbot-extension-portal></quillbot-extension-portal>

---

## [Author Response · Author response to Decision Letter 0]

24 Apr 2023

Reviewer #1

Comment: "The manuscripts was well written with just minor correction "Every year delirium complicates hospital stays for greater than 2.3 million adults of 65 years""

Response: We have added "off" to the sentence. It now reads: "Every year delirium complicates hospital stays for greater than 2.3 million adults of 65 years and older who are hospitalized in the United States[1]" (line 43).

Comment: "The number of events, i.e., the smaller (should be the small) of the number of participants with the outcome and the number of participants without the outcome,[35]."

Response: There are two numbers: (1) the number of participants with the outcome and (2) the number of participants without the outcome. Whichever is smaller is the number of events. We agree that this sentence was confusing. We have rephrased it to now read: "The number of events, i.e., the smaller number between the number of participants with the outcome and the number of participants without the outcome,[35] needed to be greater than 20 to minimize overfitting[22]" (line 412).

Comment: "I would like for the authors to elaborate on the consensus? The consensus technique employed, is it a Delphi technique? and the consensus reached by the two independent reviewer was two, a references to this should be stated to support that two independent reviewer can be used to reach a consensus."

Response: We have misused the word "consensus" in this context. We meant "agreement". We have replaced the occurrences of the word "consensus" with the word "agreement".

Reviewer #2

Comment: "ln page 9, General surgical hospitals have both medical and surgical patients wards, l am a little worried how those patients in the General Hospital wards that might have both medical and surgical patients were also excluded from the study?"

Response: We appreciate your comment on this because we did not make this exclusion criterion sufficiently clear. We excluded studies in which it was unclear what units had been included in model development and/or validation. In these cases, the study authors often used phrases such as "general hospital" patients or patients from "general hospital units". These descriptions failed to provide a clear indication about whether the unit type(s) was medical and/or surgical. We have edited this exclusion criterion to now read: "total hospital patient population (because it was unclear what unit types were included)" (lines 150-151).

Comment: "ln page 15, understand the review of the articles were those written in English, what efforts were made to get appropriate English language translational application to get some of the excluded journal articles that were excluded in the study please?"

Response: We used the interlibrary loan system that is offered by our university. We requested this article in English (original or translation), but it was unavailable. We mentioned that we had used our interlibrary loan system on page 9 in point 5 of the exclusion criteria. We have also expanded the sentence on page 15 to now read: "One study was excluded because the full text of the article was unavailable in English after a request had been made via our university interlibrary loan system" (lines 267-268).

---

## [Editor Report · Decision Letter 1]

26 Apr 2023

Risk of bias in prognostic models of hospital-induced delirium for medical-surgical units: A systematic review

PONE-D-22-28224R1

Dear Dr. Snigurska,

We’re pleased to inform you that your manuscript has been judged scientifically suitable for publication and will be formally accepted for publication once it meets all outstanding technical requirements.

Kind regards,

Blessing Onyinye Ukoha-kalu, Ph.D

Guest Editor

PLOS ONE

Additional Editor Comments (optional):

Reviewers' comments:

<quillbot-extension-portal></quillbot-extension-portal>

---

## [Editor Report · Acceptance letter]

2 May 2023

PONE-D-22-28224R1 

Risk of bias in prognostic models of hospital-induced delirium for medical-surgical units:
A systematic review 

Dear Dr. Snigurska:

I'm pleased to inform you that your manuscript has been deemed suitable for publication in PLOS ONE. Congratulations! Your manuscript is now with our production department. 

Kind regards, 

on behalf of

Dr. Blessing Onyinye Ukoha-kalu 

Guest Editor

PLOS ONE